



# Assessment of current methane emissions quantification techniques for natural gas midstream applications

Yunsong Liu[1,2], Jean-Daniel Paris[1,2], Gregoire Broquet[1], Violeta Bescós Roy[3], Tania Meixus Fernandez[3], Rasmus Andersen[4], Andrés Russu Berlanga[5], Emil Christensen[4], Yann Courtois[6], Sebastian Dominok[7], Corentin Dussenne[6], Travis Eckert[8], Andrew Finlayson[9], Aurora Fernández de la Fuente[5], Catlin Gunn[10], Ram Hashmonay[11], Juliano Grigoleto Hayashi[10], Jonathan Helmore[9], Soeren Honsel[12], Fabrizio Innocenti[9], Matti Irjala[13], Torgrim Log[14,15], Cristina Lopez[6], Francisco Cortés Martínez[5], Jonathan Martinez[16], Adrien Massardier[17], Helle Gottschalk Nygaard[4], Paula Agregan Reboredo[5], Elodie Rousset[6], Axel Scherello[12], Matthias Ulbricht[7], Damien Weidmann[10,18], Oliver Williams[10], Nigel Yarrow[9], Murès Zarea[19], Robert Ziegler[20], Jean Sciare[2], Mihalis Vrekoussis[2,21], Philippe Bousquet[1]

[1]Laboratoire des Sciences du Climat et de l'Environnement (LSCE/IPSL), CEA-CNRS-UVSQ, Université Paris-Saclay, Gif-sur-Yvette, 91191, France
[2]Climate and Atmosphere Research Center (CARE-C), the Cyprus Institute, Nicosia, 2113, Cyprus
[3]Enagás, S.A. Paseo de los Olmos, Madrid, 19 28005, Spain
[4]Dansk Gasteknisk Center a/s Dr. Neergaards Vej 5B 2970 Hørsholm
[5]SENSIA Solutions, Av. Gregorio Peces Barba, 1, Leganés, Madrid, 28919, Spain
[6]GRTgaz RICE, 1-3, rue du Commandant d'Estienne d'Orves, Villeneuve-la-Garenne, 92390, France
[7]ADLARES GmbH, Oderstraße 65, Teltow, 14513, Germany
[8]SeekOps, 1205 Sheldon Cove, TX 78753, Austin
[9]National Physical Laboratory, Hampton Road, Teddington, Middlesex, TW11 0LW, UK
[10]MIRICO, Unit 6 Zephyr Building, Eighth St, Didcot OX11 0RL, UK
[11]OPGAL, HaNapah St 11, Karmiel, Israel
[12]Open Grid Europe GmbH, Gladbecker Straße 404. Essen, 45326, Germany
[13]Aeromon Oy, Esterinportti 1, Helsinki, 00240, Finland
[14]GASSCO, Bygnesvegen 75, Kopervik, 4250, Norway
[15]Department of Safety, Chemistry and Biomedical Laboratory Sciences, Western Norway University of Applied Sciences, Haugesund, 5528, Norway
[16]Bureau Veritas Emissions Services, ZA LENFANT, 405 rue Emilien Gautier CS60401, AIX EN PROVENCE CEDEX, 13591, France
[17]Bureau Veritas Exploitation, ZA LENFANT, 405 rue Emilien Gautier CS60401, AIX EN PROVENCE CEDEX, 13591, France
[18]STFC Rutherford Appleton Laboratory, Space Science and Technology Department, Didcot, Oxfordshire, OX110QX, UK
[19]ENGIE Research & Innovation, 1 Place Samuel de Champlain, Courbevoie, 92400, France
[20]JRC Energy Institute, P.O.Box2, Petten, 1755ZG , Netherlands
[21]University of Bremen, Institute of Environmental Physics and Remote Sensing (IUP) & Center of Marine Environmental Sciences (MARUM), Bremen, D-28359, Germany

*Correspondence to*: Yunsong Liu (yunsongliu@yeah.net)

**Abstract.** Methane emissions from natural gas systems are increasingly scrutinized and accurate reporting requires site- and source-level measurement-based quantification. We evaluate the performance of ten available, state-of-the-art $CH_4$ emission quantification approaches against a blind controlled release experiment at an inerted natural gas compressor station in 2021. The experiment consisted of 17 blind, 2-hour releases at single or multiple simultaneous exhaust points. The controlled



releases covered a range of methane flow rates from 0.01 kg h$^{-1}$ to 50 kg h$^{-1}$. Measurement platforms included aircraft, drones, trucks, van, and ground-based stations, as well as handheld systems. Herewith, we compare their respective strengths, weaknesses, and potential complementarity depending on the emission rates and atmospheric conditions. Most systems were

able to quantify the releases within an order of magnitude. The level of errors from the different systems was not significantly influenced by release rates larger than 0.1 kg h$^{-1}$, with much poorer results for the 0.01 kg h$^{-1}$ release. It was found that handheld OGI cameras underestimated the emissions. In contrast, the 'site-level' systems, relying on atmospheric dispersion, tended to overestimate the emission rates. We assess the dependence of the emission quantification performance against key parameters such as wind speed, deployment constraints and measurement duration. At the low windspeeds

encountered (below 2 m s$^{-1}$), the experiments did not reveal a significant dependence on wind speed. The ability to quantify individual sources was degraded during multiple-source releases. Compliance with the Oil and Gas Methane Partnership (OGMP2.0) highest level of reporting may require a combination of the specific advantages of each measurement technique and will depend on reconciliation approaches. Self-reported uncertainties were either not available, or based on standard deviation in a series of independent realizations or fixed value from expert judgement or theoretical considerations. For most

systems, site-level overall relative errors estimated in this study are higher than self-reported uncertainties.

## 1 Introduction

Methane, a key constituent of natural gas, is a powerful short-lived (11.8 years) greenhouse gas and has about 29.8 times the global warming potential of $CO_2$ on a 100-year horizon (IPCC, 2021). Natural gas consumption has increased by 2.2% over the last decade to reach 4307.5 billion standard m$^3$, with the existing reserves reaching 188.1 trillion m$^3$ in 2021 (BP 2022).

Global demand for natural gas is projected to grow to approximately 4500 billion m$^3$ in 2030 and 5100 billion m$^3$ in 2050 (IEA, 2021). Although the combustion of natural gas releases significantly less $CO_2$ per unit of energy produced than other fossil fuels, methane emissions due to procedures leading to venting, to unintentional leaks and incomplete combustion associated with the supply chain may erode the climatic advantage of natural gas as a transition energy compared to liquid fuels if not addressed (Balcombe et al., 2017; Cooper et al., 2021; Zimmerle et al., 2020). Improving $CH_4$ emission detection

and reporting across the natural gas value chain is thus critical to understanding and mitigating the emission sources to enable a large-scale transition to natural gas.

Intensive research has recently focused on quantifying $CH_4$ emissions from different sectors of the natural gas supply chain (Bell et al., 2017; Crow et al., 2019; Duren et al., 2019; Roscioli et al., 2015; Defratyka et al., 2021; Balcombe et al., 2022). To continuously improve reporting through better quantification of emissions in natural gas value chain, different

measurement systems have been developed and applied in the field during the past decade (Allen et al., 2013; Ars et al., 2017; Johnson et al., 2021; Morales et al., 2021; Sherwin et al., 2021; Bell et al., 2020). In many cases, in natural gas production areas $CH_4$ emissions derived from atmospheric measurements were larger than the values reported in inventories at the basin scale (Harriss et al., 2015; Alvarez et al., 2018; Rutherford et al., 2021; Foulds et al., 2022), although





underestimation might not be systematic in poorly constrained production regions such as the Western Russian Arctic or Arabian Gulf gas fields (Petäjä et al., 2020; Paris et al., 2021). Inventory under-reporting has been attributed to a variety of potential reasons: reporting based on assumptions of past years' activity while activity increases; lack of accounting for all sources in emission inventories; lack of accounting for specific and time-limited venting operations; aging equipment and aging plants; challenging spatial or temporal aggregation of activities or missing specific super-emitters.

Facing this challenge of reconciling inventories with measurements and in order to monitor progress in emission reduction policies, the Oil and Gas Methane Partnership (OGMP2.0; https://www.ogmpartnership.com), as a voluntary initiative, encourages the use of site-level measurement to reconcile source- and site-level emission estimates. This approach is relevant to bridge the gap between industry practice of source-level (bottom-up) approach, and site-scale measurements (Allen et al., 2014; Olczak et al., 2022). However, measuring site scale emissions relies on a range of measurement systems, which have highly variable performance at this scale.

Controlled release experiments and intercomparison studies have been used to improve and evaluate the performance of methane emission measurement systems (e.g., Albertson et al., 2016; Feitz et al., 2018; Ravikumar et al., 2019; Edie et al., 2020; Defratyka et al., 2021b; Kumar et al., 2021; Morales et al., 2022). Ravikumar et al. (2019) reported the evaluation of the results from 10 vehicles, drones, and plane-based mobile $CH_4$ leak detection and quantification technologies through single-blind controlled release tests. They found that 6 of the 10 technologies could correctly detect over 90% of the test scenarios and correctly assign a leak to a specific equipment in at least 50% of test scenarios. Bell et al. (2020) assessed 12 $CH_4$ emission measurement technologies. They found that localization by handheld and mobile technologies is more accurate than continuous monitoring systems. However, Kumar et al. (2022) reported 20%-30% precision for the estimate of controlled $CH_4$ release rates when relying on either mobile or fixed station networks. Their localization of the releases was better when relying on fixed stations. With the rapid development of current technology, Sherwin et al. (2021) have shown that an airplane-based hyperspectral imaging $CH_4$ emission detection system can detect and quantify over 50% of total emissions from super-emitters those fewer than 20% of sources contribute more than 60% of total emissions (Duren et al., 2019). Moreover, the airborne $CH_4$ measurement technology reported by Johnson et al. (2021) can detect, locate and quantify individual sources at or below the magnitudes of recently regulated venting limits with $\pm$ 31%-68% quantification uncertainties.

These studies propose conclusions that strongly depend on the specific experimental set-up, mobile of fixed platforms, sensors, sampling strategies and applied models. With the regular improvement of instruments and techniques, new inter-comparisons based on controlled releases and involving a wide range of techniques are needed periodically. Our study aims at providing an update on the current capabilities in a scenario that replicates real conditions and at fulfilling this requirement, focusing on mature technologies available in Europe.

In the present study, we investigate the performance of various available techniques to quantify emissions in a blind-controlled release experiment. The experiment was held at a mothballed ($N_2$ inerted) compressor station of a defunct underground gas storage facility, providing a realistic environment for such measurements. It was organized by the European

Gas Research Group (GERG, https://www.gerg.eu/) in 2021. The range of emissions and the configuration of exhaust points aimed to reproduce highly realistic situations occurring in the midstream natural gas industry, including transmission

pipelines, compressors stations and storage facilities that connect upstream production to downstream distribution and end users (GIE and MARCOGAZ, 2019). The experiment included 17 blind, controlled 2-hour releases with single or multiple emission sources. The controlled releases covered a wide range of situations, such as different flow rates (from 0.01 kg h$^{-1}$ to 50 kg h$^{-1}$), release heights (ranging from 1 m to 28 m), and gas outlet shapes. In addition, the actual compressor station piping maze and equipment surrounding the release points provided the challenging air flow environment encountered in a

site in operation.

Twelve different promising measurement systems were selected to participate in this one-week campaign by GERG. The aim was to compare and evaluate these measurement systems (including mobile, ground-based, and handheld measurement platforms) to quantify $CH_4$ emissions at the industrial site level and analyze their respective strengths, weaknesses, and potential complementarity depending on the emission and atmospheric conditions. The study focuses on quantifying

emissions from single or multiple emission points. The detection and identification of those leaks are a prerequisite to this quantification but they are not evaluated in the present study as release points were known by the measurement systems' operators.

## 2 Methodology

### 2.1 Site description

A mothballed compressor station was selected as the test site. The compressor station, located in Spain, has various compression equipment for injecting and treating gas extracted from a nearby underground gas storage (Fig. 1). It is not in operation and is completely inerted with nitrogen. There are no significant natural or anthropogenic sources of methane identified in the area. The site is surrounded by flat roads from the outside and inside, making it suitable for vehicle-based mobile measurements.

Five selected gas outlet points, hereafter called nodes, were embedded in the site infrastructure. The nodes were split into two areas: Area A included Node 1, and Area B includes Nodes 2-5 (Fig. 1). Node 1 was located at the top of the vent stack of the site at the height of 28 m. In this case, the chosen exit type was open-ended to simulate the emission conditions in vent stacks. Node 2 was 9 m above ground level with an open-ended exit. Node 3 was 4 m high with openings in a ring-shaped small pipe. Node 4 was a linear tube three meters long with holes along it at 1.5 m height. Node 5 was an open outlet at 1.5

m height, dedicated exclusively to the tests with the lowest emission rates.



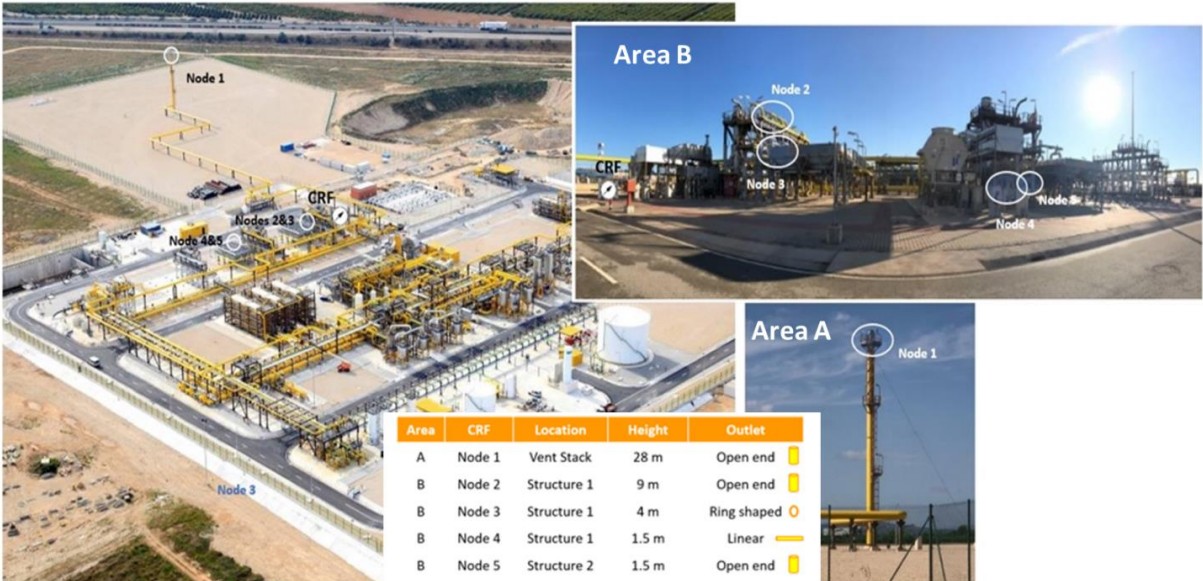

**Figure 1: Aerial view of the Enagas site in Spain and node location (white circles). The controlled release facility is indicated by a white disk marked "CRF".**

## 2.2 Controlled release facility

The controlled release facility (CRF) is a portable flow control system purposefully designed and configured by the National Physical Laboratory (NPL) to create 'real-world' gaseous emission scenarios. A detailed description can be found in Gardiner et al. (2017). The system (Fig. 2) enables the operator to replicate a variety of gaseous emissions at comparable scales in a range of industrial settings to validate emissions monitoring methodologies under field conditions. The facility is computer-controlled and monitored, allowing for the execution of pre-written operational programs and the analysis of flow

data post-test. Communication to the instrument is made via a low-voltage umbilical cable, allowing the operator to control the system from a distance of up to 50 m from the gas blending equipment. The so-called 'MidiCRF' system was used when the flow was below 1.2 kg h$^{-1}$ (Node 5). Its principle derives from a simplified version of the CRF. The uncertainties of the CRF and MidiCRF are dominated by the calibration uncertainty. Calibration was performed on site, prior to tests commencing, with the same source gas as used in the tests. The systems were calibrated by NPL using volumetric piston

based calibrators (Mesa Labs) with measurement traceability to national standards.

NPL provided operational training to Enagas staff, who then operated the CRF for the execution of the tests.



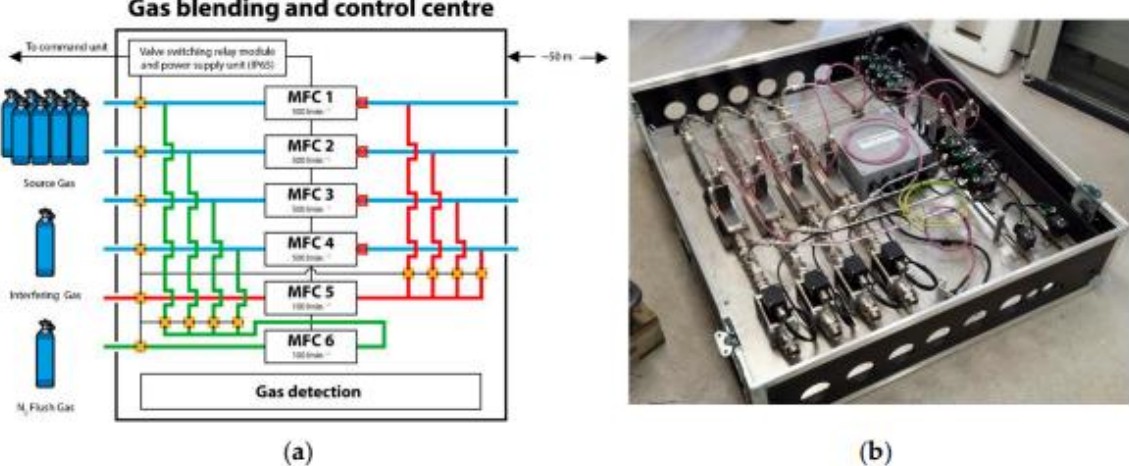

**Figure 2: (a) the Controlled Release Facility schematic and (b) photograph of the flow control system (Gardiner et al., 2017).**

### 2.3 Test scenarios and organization of the experiment

The 17 releases were performed from October 4 to October 8, 2021. They covered a range of situations combining different total flow rates (0.01 kg h$^{-1}$-50 kg h$^{-1}$), across single or multiple nodes. This approach aimed at simulating a variety of fugitive and venting emissions in natural gas midstream sites.

The releases involved single or multiple nodes with a constant emission rate over 2 hours. The releases were "blind", i.e., the release rates were not known by the participants. The series of release rates were established in advance and ordered

randomly within the range of 0.01 kg h$^{-1}$ to 50 kg h$^{-1}$ (Table 2). Two releases took place in Area A only, 14 in Area B only, and one in both Areas A and B. The participants knew the areas of emission (A and/or B) but not the exact emitting node(s) in the case of Area B. Participants also knew the range of possible emission rates and the timing of the releases. Participants did not know each other's results until after all participants blindly uploaded their results to an 'upload only' server, three weeks after the end of the campaign week.

The lowest release rates (below 0.5 kg h$^{-1}$) were dedicated to evaluating the quantification limit, defined here as the lower limit below which a technique does not provide relevant emission estimates.

Due to the linearity of the CH$_4$ atmospheric dispersion, we hypothesize that across the releases, and a threshold will emerge between a "low concentration regime" (where measured concentrations are commensurate with instrumental accuracies) or a "high concentration regime" (where instrumental accuracies of methane measurements become negligible against

methodological approaches or ancillary measurements). This distinction aims to be generic and may not describe the behavior of a particular instrument or method. However, in a low-concentration regime, the relative uncertainties in emission rate estimates are expected to decrease with increasing release rates. The quantification limit should thus correspond to the emission threshold above which the measurement uncertainties are sufficiently low so that the uncertainty in the emission estimate does not depend on the release rates. We used a relatively uniform sampling of the emission rates to infer it.



Within each 2 h release, the series of measurements by the different participants were sequenced to minimize the impact of a specific quantification system on others. For example, drones flew sequentially to avoid any collision risk. The helicopter flew over the site only at the very end of the releases to avoid disturbing plume dispersion for other groups. The drones generate turbulences that can influence the structures of the plume measured by other platforms (in particular by Lidar 2), which can perturb the corresponding emission computation. An initial organizational briefing ensured the alignment of all

technology providers and a smooth succession of releases and measurements. During the campaign week, permanent coordination by radio was applied between site coordinators and all involved groups. Experiment details and sequencing technologies were shared with all participants through a paperboard on the site. The different quantification systems relied on different measurement durations to provide release estimates due to this organization but also because they followed different operating protocols.

A sonic anemometer (Vaisala WXT530) attached to a mast was located between Areas A and B at 5 m height to perform wind measurements during the campaign.

The Drone 2 group performed daily background measurements prior to any release (using drone-based optical $CH_4$ measurements). Morning daily concentrations remained within 2.2-2.5 ppm (reported range). These background measurements suggested that there was no large local $CH_4$ source near the site. Therefore, it is unlikely that any significant

$CH_4$ enhancement from outside the site may have influenced the release experiment.

**Table 1: Test scenarios with detail of emission rates per node (unit in kg h$^{-1}$). The reported uncertainties are based on 2 standard deviations, providing a confidence interval of 95%.**

| Test | Total emission rate | Node1 | Node2 | Node3 | Node4 | Node5 |
|------|---------------------|-------|-------|-------|-------|-------|
| 1 | 2.6 ±1.8 | | | 2.6 ±1.8 | | |
| 2 | 5.7 ±0.7 | 5.7 ±0.7 | | | | |
| 3 | 1.2 ±0.01 | | | | | 1.2 ±0.01 |
| 4 | 22.7 ±2.2 | | 9.7 ±0.5 | 3.1 ±1.8 | 10.0 ±1.2 | |
| 5 | 5.7 ±1.3 | | 2.0 ±0.5 | | 3.6 ±1.2 | |
| 6 | 22.4 ±2.2 | | 9.8 ±0.5 | 2.7 ±1.8 | 10.0 ±1.2 | |
| 7 | 18.9 ±0.7 | 18.9 ±0.7 | | | | |
| 8 | 46.4±2.3 | 11.0 ±0.7 | 15.2 ±0.5 | 9.0 ±1.8 | 11.1 ±1.2 | |
| 9 | 0.1 ±0.0001 | | | | | 0.1 ±0.0001 |
| 10 | 5.1 ±1.2 | | | | 5.1 ±1.2 | |
| 11 | 8.1 ±1.2 | | | | 8.1 ±1.2 | |
| 12 | 32.5 ±2.2 | | 16.7 ±0.5 | 5.9 ±1.8 | 9.9 ±1.2 | |
| 13 | 0.5 ±0.01 | | | | | 0.5 ±0.01 |
| 14 | 7.03 ±1.31 | | 2.5 ±0.5 | | 4.5 ±1.2 | |
| 15 | 0.01 ±0.0001 | | | | | 0.01 ±0.0001 |
| 16 | 3.8 ±1.2 | | | | 3.8 ±1.2 | |
| 17 | 14.6 ±2.2 | | 2.3 ±0.5 | 9.8 ±1.8 | 2.5 ±1.2 | |





## 2.4 Participants and measurement systems

Twelve quantification systems were selected based on an internal review by the GERG consortium (GERG, 2021, "Technology Benchmark for site-level methane emissions quantification"-Phase I-GERG, https://www.gerg.eu/projects/methane-emissions/gerg-technology-benchmark-for-site-level-methane-emissions-quantification-phase-ii-a/). The ability to detect leaks was not part of the criteria as the present study focuses on quantification. Besides the performance of each measurement system, the criteria included high Technology Readiness Level

(TRL), demonstrated ability to perform such measurements on-site, and the possibility for the service to be performed commercially by an independent operator. Table 2 summarizes the main characteristics of these systems. These quantification systems combine measurement platforms, instruments, and post-processing algorithms to derive emission rates. The systems are based on handheld, vehicle, drone, and airborne mobile platforms and ground-based fixed measurements. The measurement devices include optical gas imaging cameras, DIAL lidar, off-axis integrated cavity output

spectroscopy, and tunable diode laser spectrometry, as well as an early prototype direct quantification device, a Venturi effect High Flow Sampling system, designated subsequently by Hi-Flow.

Each system implemented its own quantification methodologies and associated developed quantification software to derive $CH_4$ emission rates. These approaches include inverse dispersion modeling, mass balance, tracer ratio and other proprietary data algorithms. The reporting of the emission rates was done according to a specific template.

The self-reporting of uncertainties, however, was not mandatory and no specific reporting format was required. Six of the systems provided their diagnostics of uncertainties in the estimates (hereafter all uncertainties are provided in terms of 1-sigma values). Lidar 2 reported expanded uncertainty providing a 95% level of confidence.

A company operating two systems, one drone based and another one car-based, withdrew from the experiment and did not report their data. They reported that their measurement protocol was to be optimized.

In addition to these relatively mature technologies, a direct quantification equipment for fugitives was able included in the tests. The equipment is a hand-held device that used a venture tube supplied by a compressed air cylinder (Hi-Flow). This equipment is a prototype with a low TRL, and it was included in the test to assess its performance for fugitives' quantification.





**Table 2: A summary of the systems participating in the campaign.**

| Name | Platform | Sensor | Quantification algorithm | Assessment type |
|---|---|---|---|---|
| Drone 1 | Matrice 300 RTK from DJI | Tunable Diode Laser Spectrometry | Reverse dispersion modeling, considering the location of the plume, sensor measurements and local weather data | Site-level |
| Lidar 1 | Helicopter MD-900 | LiDAR DIAL | Direct estimation by multiplying the integrated gas concentration, the respective wind speed and the sine of the angle between the fence line and wind direction | Site-level |
| Tracer | Van | Off-axis integrated cavity output spectroscopy | Calculated as the integrated signal of $CH_4$ concentration relative to the integrated signal of tracer gas concentration | Site-level |
| Lidar 2 | Truck | Differential absorption lidar (DIAL) | Determined by combining the concentration map with wind speed and direction | Site-level |
| Drone 2 | DJI M300 UAS | An in-situ tunable diode laser absorption spectrometer | Proprietary data algorithms based on an engineering control volume model | Site-level |
| Fixed 1 | Ground | Laser dispersion spectroscopy operating in the midIR region | The algorithm combines gas concentration data of each retroreflector with meteorological data | Source-level |
| Fixed 2 | Unmanned cameras | Two OGI cameras: an uncooled LWIR detector and a cooled MWIR detector | Depends on three variables: thermal contrast between the plume and the background; column density; absorption peak of the target gas | Source-level |
| Hi-Flow | Handheld | A venturi tube driven by a compressed air cylinder | Determined by the gas concentration and the suction flow rate of the venturi | Source-level |
| OGI 1 | Handheld camera | Optical gas imaging (OGI) camera | Quantification software | Source-level |
| OGI 2 | Handheld camera | OGI camera | Quantification software | Source-level |



## 3 Data collection and analysis

The primary purpose of the experiments was to assess the ability to infer the total methane emission rate during each release.

Therefore, the reporting focused on providing a total emission estimate for each release. During multiple-node releases, we also considered detailed reported estimates for individual nodes when available from the participants. The ability to provide estimates per individual source during a multiple release was considered a desirable feature of site-level quantification techniques.

As a normalized performance indicator, the absolute value of the relative error (called hereafter "absolute error", $|(E_{estimate}-E_{real})/E_{real}|$) was computed for each release and each provider. $E_{estimate}$ is the estimate provided by a given participant, and $E_{real}$

is the actual emission rate). The distributions of absolute error are analyzed per release (considering each provider as a single realization) or per provider (considering each release as a different realization).

Table 3 gives an overview of the number of results provided by each participant. It indicates, for each experiment, whether a given participant provided the estimate for the total emission rate, partial emission rate estimates where one or several nodes

may be missing, or an estimate that is not valid. Overall, there is no single release that was reported by all participants, and no system reported all releases; the number of total emission estimates was between 5 and 9 for a given release, and between 5 and 16 for a given system. The Hi-Flow prototype was not authorized to work on Nodes 1 and 2, as those nodes were considered risky (difficult to access) for the operator. The limited amount of data reported directly constrained our ability to identify robust statistical relationships between the errors in the release rate estimates and potential drivers of the

quantification such as the meteorological conditions or the type of $CH_4$ releases.

All participants followed their own process to provide quality control and validate their estimates. Some participants excluded data points considered poor and provided reduced coverage of the releases prioritizing lower uncertainty, while others provided extensive coverage. Each data provider relied on its own judgment and procedures to balance the quantity and quality of the estimates. This balance is essential to consider when evaluating the respective merits of each system as a

high overall precision may be a trade-off with a high "coverage" of the release rates. It should be noted that in real-life operations, less stringent time constraints may apply and more time may be available on site compared to the 2h per release of the present study. Each provider reported specific limitations and challenges explaining the coverage of the releases after the campaign.







**Table 3: Overview of valid emission estimates for each release, including the 0.01 and 0.1 kg h⁻¹ releases. The letter indicates the availability of estimation. T: total emissions were captured; P: partial emission rate, one or several nodes may be missing from the total; 0: no estimate or value considered invalid by the provider. In the case of Lidar 2, some results were considered poor due to the influence of drones during the campaign.**

| Release ID | 1 | 2 | 3 | 4 | 5 | 6 | 7 | 8 | 9 | 10 | 11 | 12 | 13 | 14 | 15 | 16 | 17 | Nb full | % Full |
|---|---|---|---|---|---|---|---|---|---|---|---|---|---|---|---|---|---|---|---|
| Drone 1 | 0 | T | T | T | T | 0 | T | T | T | T | T | 0 | T | 0 | T | T | 0 | 12 | 71% |
| Lidar 1 | T | T | 0 | T | T | 0 | T | T | T | T | T | T | T | T | T | T | T | 15 | 88% |
| Tracer | T | 0 | T | T | T | T | T | T | T | T | T | 0 | T | T | 0 | T | T | 14 | 82% |
| Hi-Flow | 0 | 0 | T | P | P | P | 0 | 0 | T | 0 | 0 | 0 | T | P | T | T | 0 | 5 | 29% |
| Fixed 1 | T | 0 | T | T | T | T | 0 | P | T | T | T | T | T | T | T | T | T | 14 | 82% |
| Lidar 2 | 0 | T | 0 | 0 | T | T | T | P | 0 | 0 | 0 | 0 | T | T | 0 | 0 | T | 7 | 41% |
| OGI 1 | T | T | T | T | T | T | T | T | T | T | T | T | T | T | T | T | P | 16 | 94% |
| Drone 2 | T | T | T | T | P | T | T | T | T | T | T | T | T | T | T | T | T | 16 | 94% |
| Fixed 2 | T | T | 0 | P | P | T | T | P | 0 | T | T | T | 0 | P | 0 | T | T | 9 | 53% |
| OGI 2 | T | T | T | P | T | P | T | P | T | T | T | T | T | T | T | T | T | 14 | 82% |
| Nb full estimates | 7 | 7 | 7 | 6 | 8 | 6 | 8 | 5 | 8 | 8 | 8 | 6 | 9 | 7 | 7 | 9 | 7 | | |

## 4 Results and discussion

### 4.1 Qualitative assessment of the total emission estimates per participant

In line with quantifying total site emissions, Figure 3 compares the total emission estimates provided by each participant with actual total emission rates per release. It displays linear regressions between the estimated and actual emission rates (without weighting the estimates based on the diagnostics of uncertainties).

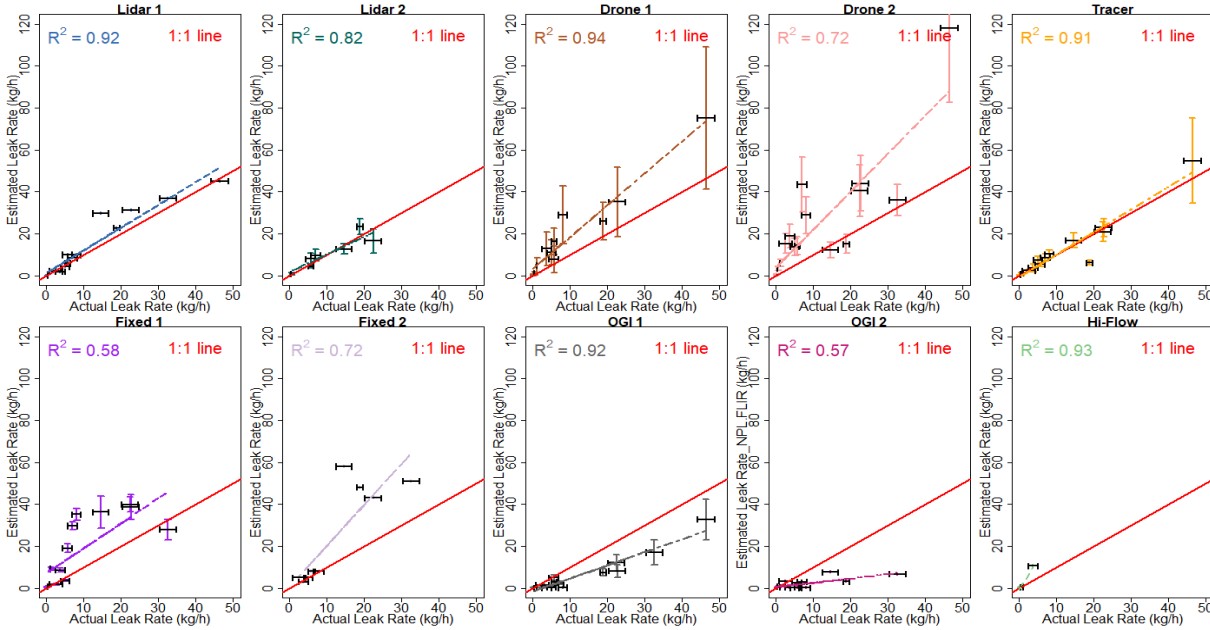





**Figure 3: Scatter plot of estimated and actual rates for the releases for each participant. Linear regression (dashed line) including the 1:1 line (red) shown for reference. The horizontal uncertainty bars are the 1-sigma uncertainties of the controlled release facility. The vertical error bars are uncertainties provided by the participant.**

Regarding biases on estimates, Lidar 1 slightly overestimated the emission rates, especially in the middle range of release rates. There was no significant bias in the release estimates from Lidar 2, limited to 22.4 kg h$^{-1}$. Drone 1 and Drone 2 tended to overestimate emission rates. All estimates but one from Tracer bore errors that fit in the 1-sigma uncertainty specified for this system. Fixed 1 tended to overestimate the emission rates from 5 kg h$^{-1}$ to 30 kg h$^{-1}$. For Fixed 2, the performance was better for lower emissions (below 10 kg h$^{-1}$) and tended to overestimate the emission rates above 10 kg h$^{-1}$. By contrast, OGI 1 and OGI 2 tended to underestimate the emission rates, likely influenced by cases where the distance to node could not be within the recommended range. There was no obvious bias for Hi-Flow, but it provided only estimates for three single-node release in the lower rate range: 0.5-3.8 kg h$^{-1}$.

In summary, the quantification systems of Lidar 1, drones and both fixed sensors generally overestimated the emission rates (with regression slopes ranging from 1.08 for Lidar 1 to 1.96 for Fixed 2), and the systems of handheld OGI generally underestimated them. However, two site-level systems did not follow this trend. The estimates from Tracer and Lidar 2 were close to the actual rates. In the present study, the number of results provided by Lidar 2 is small to assess any biases since the release they could not measure the appropriate nodes were excluded from the analysis.

## 4.2 Total release emission estimates: quantitative synthesis

Figure 4 provides the distribution of absolute error for the series of estimates from each participant, excluding the results for the two smallest releases of 0.01 kg h$^{-1}$ and 0.1 kg h$^{-1}$, whose specific goal was to assess the quantification limits.

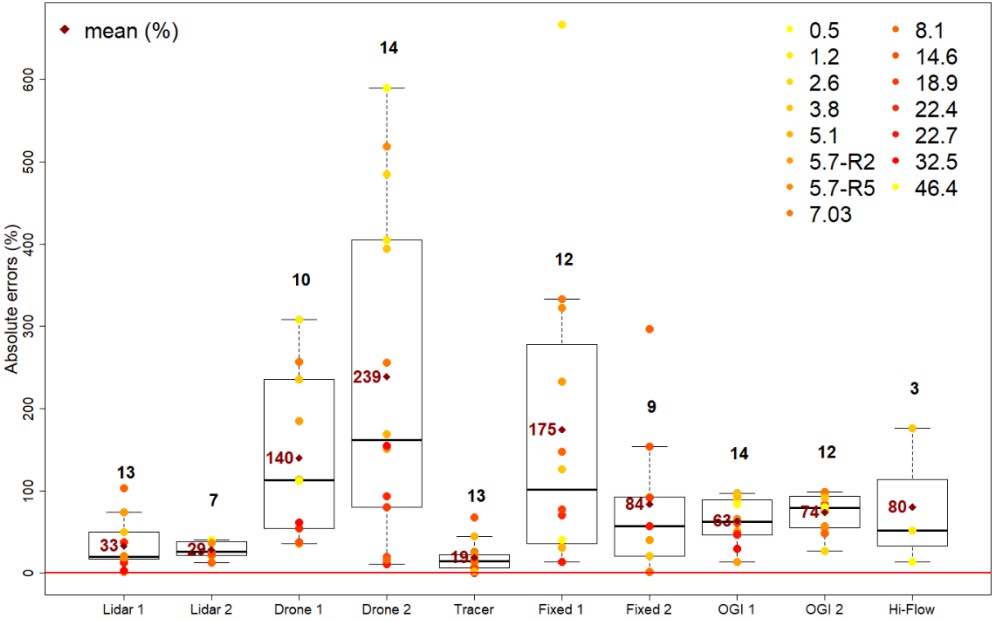



**Figure 4: Absolute errors for each system, in percent. The color scale corresponds to the actual rates of the different releases, given in the top right corner (kg h$^{-1}$). Whisker plots indicate the median, interquartile range, min, and max (excluding outliers) of the distributions. The average values are also indicated (dark red dot, %). The number of points accounted for in the statistical distribution is indicated on top of each whisker plot. The quantification technologies are ordered from site-level (Lidar 1, Lidar 2, Drone 1, Drone 2 and Tracer) on the left to source-level systems (Fixed 1, Fixed 2, OGIs and Hi-Flow) on the right.**

The absolute errors range from 0% to 600%, even when excluding the releases below 0.5 kg h$^{-1}$. There is a large spread of typical errors in the results from one participant to the other, with average absolute errors per participant ranging from 19% (for Tracer) to 239% (for Drone 2). Among the site-level quantification systems, Lidars and Tracer provide estimates with absolute errors typically below 50%, while estimates from both drones generally bear average absolute errors in excess of 100% errors. Fixed sensors provide intermediate performance, with an average absolute error of 84% to 175%. Low wind speeds (below 2 m s$^{-1}$) combined with short time window (20 min) to collect data may have challenged the modeling of the dispersion for the processing of drone measurements and secondary, possibly turbulence caused by the drones, which could explain the drones' high errors during the experiment. The different source-level quantification systems provide relatively consistent performance with lower 63% to 80% average absolute errors and absolute errors for any release that generally lie below 100%. Hi-Flow not commercially available, which relies on a particular sampling principle, provides good performance among the source-level measurements but only on three complete release estimates due to deployment limitations, while OGI 1 and OGI 2 provide 14 and 12 estimates, respectively. Figure S1 shows how often the estimates from a system fall within a multiplicative range of the actual values, either between half and twice the actual value or within a tenth and ten times the actual value. Notably, it highlights that some systems provide results with occasional discrepancies of more than one order of magnitude (OGI). The table excludes the releases of 0.01 kg h$^{-1}$ and 0.1 kg h$^{-1}$. The only system that limits 100% of its total release estimates within a factor of 2 (range 0.5-2) uncertainty band has the second-lowest coverage rate (Lidar 2). Conversely, OGI 1 and Drone 2 are within the factor 2 range only for 36% of the releases. Given its relevance for reconciliation, i.e., ensuring that there is no missing emission, we chose the total emissions from Table 3 to establish these statistics.

## 4.3 Parameters influencing total release estimates

### 4.3.1 Role of measurement duration

Measurement duration is one of the factors that influence deployment, with the mobilization and demobilization time. These durations were established by the technology suppliers, and are supposed to optimize results. The influence of measurement duration is expected to be the result of two competing effects: 1) integrating more data leads to a decrease in measurement error and 2) wind (and hence plume position) variation over time may 'blur' the data. Here we find that the errors are not correlated with the time used by the different participants to make measurements. Both fixed sensors (Fixed 1 and Fixed 2) integrate measurements over two hours, while helicopter-based Lidar 1 relies on nearly instantaneous images of the concentration field, with total survey duration measured in seconds, not even minutes. Tracer records on average 2.5 min per





measurement leg. To ensure an appropriate uncertainty, 20 measurements were conducted per release (in total 45 min
325 measurements). Drones had 20 minutes to cover the measurement. Suggestion to elaborate on this aspect, see comment.
While the tests analyzed here didn't perform sensitivity analysis of measurement duration for each technology, future
experiments could support the investigation of how the performance of some techniques relying on integration over
measurement durations that can vary would improve with increasing duration. At the site level, for some techniques,
increasing time coverage can also improve the ability to cover all nodes. For others, this can lead to an overestimation if not
330 considered in the estimation process.

### 4.3.2 Dependence of error on emission rate

Figure 5 shows the relation of absolute errors with the total emission rate for all 17 releases. The estimates of the 0.01 kg h⁻¹
bear errors systematically larger than 100% (100% for Lidar 1) for all participants' systems and often reach more than 500%
(3300% for Fixed 1). Therefore, 0.01 kg h⁻¹ could not be quantified with any of these systems. The wind conditions during
335 this smallest release did not appear to be more challenging than during the other releases (wind speed was 2.6 m s⁻¹; see
Section 4.3.3). Therefore, the low magnitude of this release challenged all types of systems. This finding shows that this leak
rate is below the quantification limit for most techniques. Besides this case, and even considering the 0.1 kg h⁻¹ release, the
range of errors does not appear to decrease with increasing release rates, consistent with expectations in a high-concentration
regime. Thus, the quantification limit for most systems (Tracer, Drone 1, Drone 2, Fixed 1, OGI 1, OGI 2, and Hi-Flow)
340 appears to lie between 0.01 kg h⁻¹ and 0.1 kg h⁻¹.

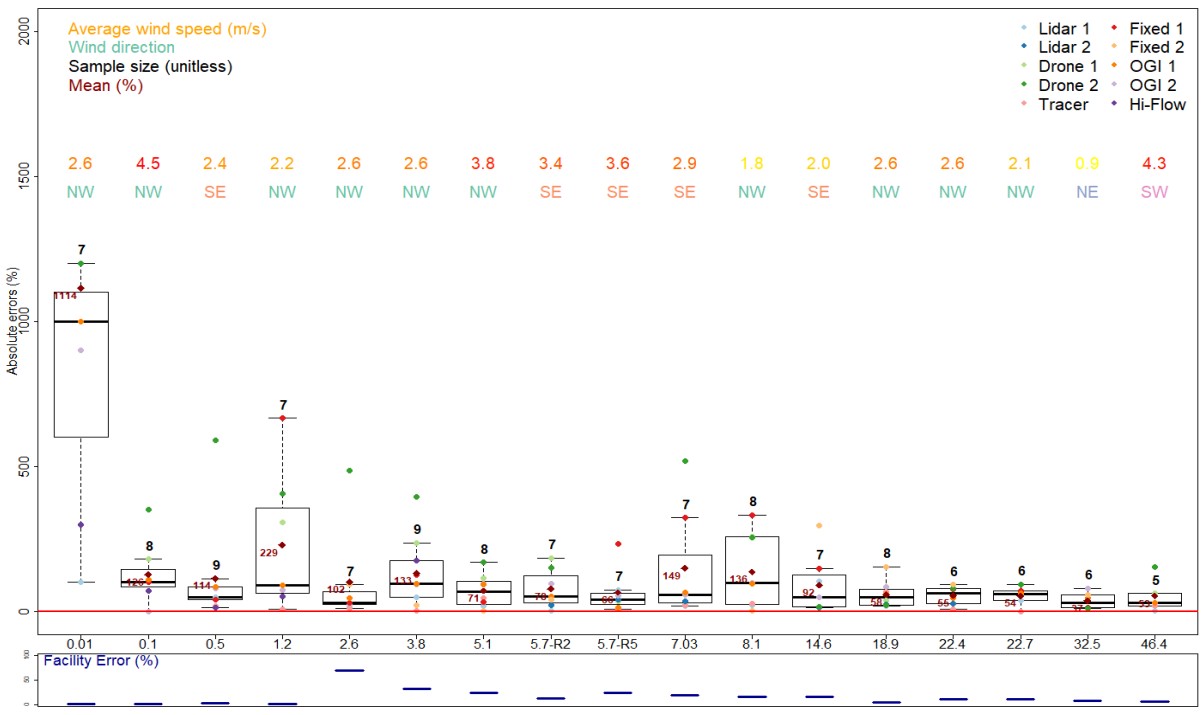





**Figure 5: Aggregated absolute errors as a function of total release rate. Dot colors correspond to individual participants (top right legend). Whisker plots indicate the median, interquartile range, min, and max (excluding outliers) for each release. The average values are also indicated. Average wind speed and direction per release are provided in the upper part of the top panel. The lower panel shows the uncertainty in the CRF rate for each release.**

### 4.3.3 Role of wind

The amplitude of the signal and the accuracy of the modeling frameworks are expected to depend strongly on the wind and turbulence conditions, primarily on the wind speed. Low wind speed (below 2 m s$^{-1}$) can be challenging for those participants relying on atmospheric dispersion models for the quantification of the emission rates (Wilson et al., 1976). The wind direction likely plays a role since the positioning of the sensors is constrained by logistical issues, due to the potential overlapping or divergence of plumes from different nodes and since some directions drive the plume against or close to obstacles impacting the atmospheric flows. Low wind speed values (below 2 m s$^{-1}$) and specific wind direction sectors prevented some participants from providing valid estimates during specific releases. However, considering the valid estimates, the results do not reveal any clear relationship between the wind speed or direction and the errors. Among the releases for which the errors were significantly larger than for the others is that of 1.2 kg h$^{-1}$ from Node 5 only and that of 8.1 kg h$^{-1}$ from Node 4 only, both at 1.5 m height in a congested area. In these cases, the average wind speeds were relatively small, and the wind was blowing from the NW. Only two other releases above 0.1 kg h$^{-1}$ were performed from Node 4 and Node 5, with NW and stronger winds and lower scatter. Better performance were reported for other releases, which were conducted under weaker and/or NW wind. A more thorough examination of individual releases with a high spread in performance is required. Overall, it should be investigated further that improving wind measurement protocols and upwind/downwind congestion characterization may lead to enhanced accuracy for the leak rate estimates.

### 4.3.4 Sensitivity to the different types of nodes

In this section, we investigate the influence of specific nodes (with a specific shape, configuration and/or location; see Section 2.1) on the relative errors. Mean absolute errors for single node releases from Node 1, Node 3, Node 4 and Node 5 are 68%, 102%, 113% and 172%, respectively (Table 4). There was no single release from Node 2 only. Node 5 bears larger absolute errors than other nodes. This might be explained by the dedication of Node 5 to the lowest rates and its proximity to the ground. This position may induce a dispersion that is more complicated to capture.

Some nodes may raise specific issues during multiple node releases e.g., because they are away from the others and thus require extensive sampling (which is notably the case for Node 1). This ability to perform extensive measurements can be considered a good discriminant of site-level techniques. However, we have only seven multiple-node releases, which systematically include Nodes 2 and 4 and exclude Node 5. This limits our ability to get robust conclusions regarding the impact of specific nodes during multiple node releases.



**Table 4: Distributions of the mean absolute errors (%) across the available, total release estimates from the different measurement systems for each release.**

| Nodes | Emission rate (kg h$^{-1}$) | Release ID | Mean absolute errors (participants) |
|---|---|---|---|
| 1 | 5.7 | 2 | 78 (7) |
| 1 | 18.9 | 7 | 58 (8) |
| 3 | 2.6 | 1 | 102 (7) |
| 4 | 3.8 | 16 | 133 (9) |
| 4 | 5.1 | 10 | 71 (8) |
| 4 | 8.1 | 11 | 136 (8) |
| 5 | 0.5 | 13 | 114 (9) |
| 5 | 1.2 | 3 | 229 (7) |
| 2&4 | 5.7 | 5 | 66 (7) |
| 2&4 | 7.03 | 14 | 148 (7) |
| 2&3&4 | 14.6 | 17 | 92 (7) |
| 2&3&4 | 22.4 | 6 | 55 (6) |
| 2&3&4 | 22.7 | 4 | 54 (6) |
| 2&3&4 | 32.5 | 12 | 37 (6) |
| 1&2&3&4 | 46.7 | 8 | 53 (5) |

Node 1 (the vent stack) is away from the other nodes and raises specific challenges for some systems. In particular, Tracer, Fixed 1, and Hi-Flow could not measure Node 1 due to accessibility issues. Its height exceeded the maximum distance of the operating range for OGI 2. Statistics for single-node releases (Table 4) showed that the results for releases from Node 1 are

better than for other single-node releases. Table 5 details the results per measurement system for the releases with emissions from Node 1 only. For those releases, Lidar 1 and Lidar 2 provide estimates with less than 25% absolute errors, and other systems can yield more than 50% absolute errors.

Figure 6 is similar to Fig. 4 but it excludes all the releases that include Node 1 (i.e., excluding releases #2, #7 and #8), and shows that the best performing technologies are Tracer, followed by Lidar 2 (limited coverage) and Lidar 1 Removing these

three releases increases the mean errors of most techniques. These results indicate that emissions from Node 1 are easier to quantify than other nodes, likely due the lack of obstacles to air flow. This is put in balance of the challenge represented by measuring at this height for source-level.






**Table 5: Absolute errors (%) for releases from Node 1 (the vent stack).**

| Release ID | R2 (5.7 kg h⁻¹) | R7 (18.9 kg h⁻¹) |
|---|---|---|
| Lidar 1 | 2 | 20 |
| Lidar 2 | 21 | 23 |
| Drone 1 | 185 | 37 |
| Drone 2 | 151 | 20 |
| Fixed 2 | 40 | 154 |
| OGI 1 | 51 | 61 |
| OGI 2 | 95 | 84 |
| Tracer | N/A | 68 |
| Mean (%) | 95 | 65 |

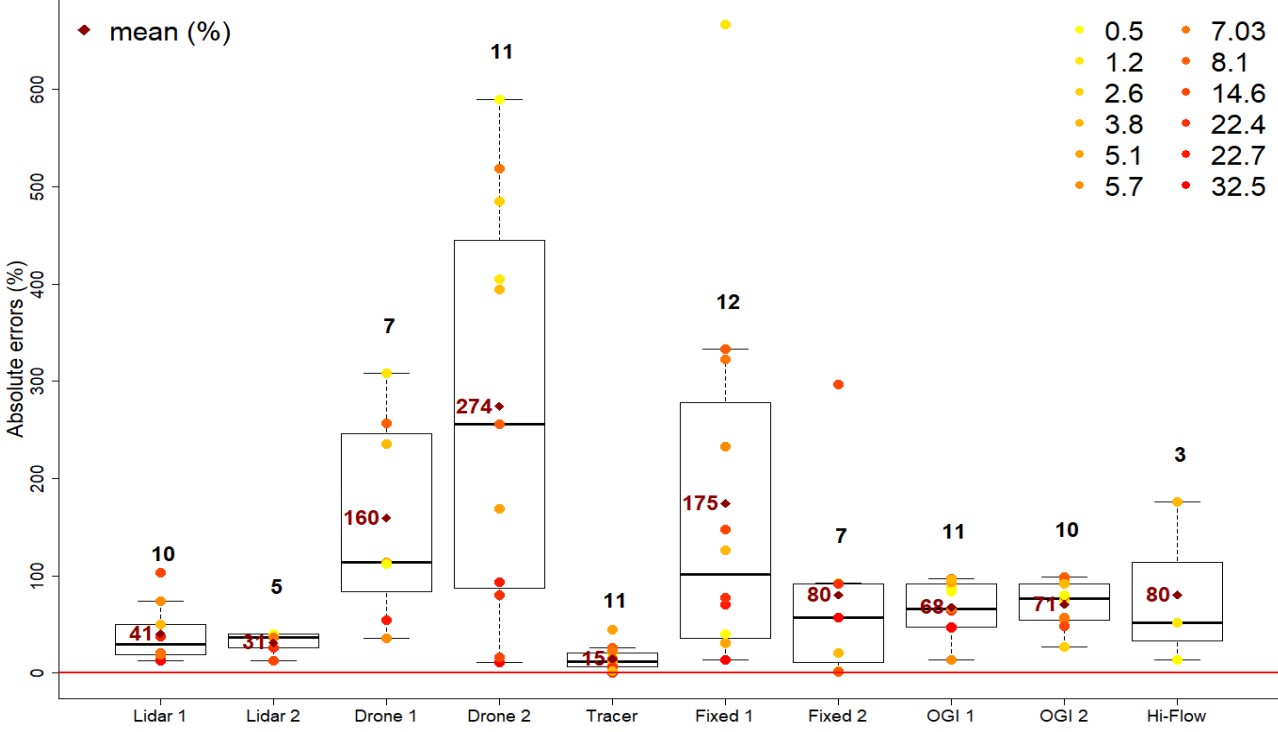


**Figure 6: Same as Figure 4 but excluding vent stack emissions.**



### 4.3.5 Are site-level performance better during single-node releases?

Ignoring the releases of 0.01 kg h⁻¹ and 0.1 kg h⁻¹, we have eight single-node releases and seven multiple-node releases, as shown in Table 1. In general, most measurement systems' total estimates of multiple-node releases are better than their

estimates of single-node releases (6 out of 9 systems, by 70% on average, as shown in Table 6). This result is unexpected since, in principle, it is more challenging to sample and properly analyze information on multiple more or less overlapping plumes arising from more or less distant sources rather than to sample and analyze a single plume from a single source. It is unclear whether this result is statistically robust or if the number of data points is too limited to raise a robust conclusion on this topic. The result is reassuring, as the purpose of site-level systems is to check whether any emission source may have

been missed through the source-level measurements. For some specific techniques (Lidars and Fixed 2), the opposite is true: single-node estimates are more accurate than their total estimates for multiple-node releases.

**Table 6: Distributions of the mean absolute errors (%) of each measurement system of single-node releases, multiple-node releases, and all releases according to the estimates provided by each participant.**

| Release Type | Single node (%) | Number | Multiple nodes (%) | Number | ALL (%) |
|---|---|---|---|---|---|
| Lidar 1 | 26 | 7 | 42 | 6 | 33 |
| Lidar 2 | 28 | 3 | 29 | 4 | 29 |
| Drone 1 | 178 | 7 | 51 | 3 | 140 |
| Drone 2 | 309 | 8 | 146 | 6 | 239 |
| Tracer | 26 | 7 | 11 | 6 | 19 |
| Fixed 1 | 205 | 6 | 144 | 6 | 175 |
| Fixed 2 | 52 | 6 | 149 | 2 | 84 |
| OGI 1 | 77 | 8 | 45 | 6 | 63 |
| OGI 2 | 81 | 8 | 60 | 4 | 74 |
| Hi-Flow | 80 | 3 | NA | NA | 80 |
| Mean (%) | 106 | N/A | 75 | NA | 94 |

### 4.4 From site level to source level: Node-level performance

This section aims at assessing the potential for mapping and attributing the site-level emissions to different sources (in complement to quantifying the total emissions) in an industrial site, focusing on individual nodes. Such single-node estimates were optionally provided during multiple-node releases by some of the measurement systems, which have the capability to distinguish the signal from the different nodes. In principle, this is a defining feature of source-level systems. However, most site-level techniques had this ability as well. The accessibility of nodes and their location nearby other leaks

have conditioned the provision of valid data by participants.





Figure 7 compares the collective performance of all techniques at the single node level during single- and multiple-node releases for all the measurement systems. Multiple-node releases were available for all nodes, excluding Node 5. Single-node releases were unavailable from Node 2. The quantification systems perform better on average when no other node emits. Node 3 is quantified with a 51% mean error when emitting alone, against 127% when part of a multiple node release.

Similarly, for Node 4, the mean error during the single-node releases is 100%, increasing to 124% during multiple-node releases. This effect is less obvious for Node 1 (comparing 67% and 71% errors). The generally better performance in quantifying individual nodes when no other node emits is likely linked to the influence of the signal from other emissions in quantifying the individual node. Source-level techniques perform equally well for individual nodes during single- and multiple-node releases. This highlights that good performance in site-level emission quantification does not necessarily

imply good performance in individual source-level quantification and that requirements for leak quantification need to be carefully specified prior to selecting a particular technique.

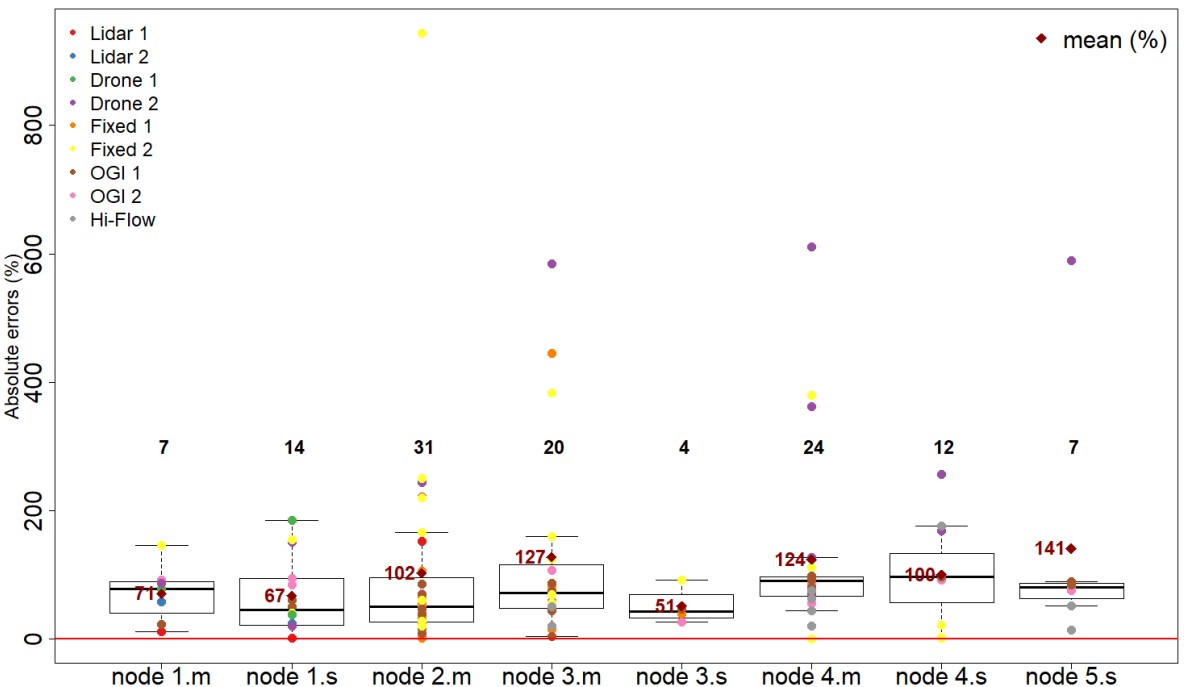

**Figure 7: Comparison of absolute errors for single node releases ("s") and multiple node releases ("m"). Excluding the releases of 0.1 kg h⁻¹ or less. Each point is a participant's node-level estimate.**

Focusing now only on single-node estimates during multiple-node releases, absolute errors on individual Nodes 1-4 are 71%, 102%, 127% and 124% respectively. These significantly higher uncertainties for Nodes 2-4 are linked to the fact that they are located in Area B, embedded in a large structure and with a close location of nodes. It could also mean that Node 1 was not accessible to less accurate techniques, creating an artificial favorable bias for Node 1. This directly impacts Nodes 2-4



uncertainties, with the possibility to combine with a possible overlap in plume dispersion if wind runs parallel to the alignment of these nodes.

Differences in node-level errors during multiple-node releases across Nodes 2-4 are not statistically significant. There is, therefore, no obvious detectable influence of the node shape on the performance in the context of multiple-node releases.

The ability of distinguishing individual nodes, even with a slightly degraded performance, is a desirable feature for site-level methods to facilitate reconciliation and verification that all sources are accounted for by source-level methods. This may be a criterion for trade-off between accuracy and ability to resolve individual sources for a given facility.

Each measurement system had its specific performance for specific combinations of nodes. Fixed 2 showed relatively larger absolute errors for Area A, and Drone 2 showed relatively larger absolute errors for Area B. OGI measurement systems showed relatively stable and smaller absolute errors than other systems during multiple node releases. In the case of Hi-Flow, its ability to characterize correctly leaks from only certain nodes is aligned with its specifications.

Unsurprisingly, source-level systems are not systematically able to capture all emissions during multiple-node releases due to constraints such as the node configuration, wind speed and wind direction. High altitude sources like Node 1 may be out of reach for source-level quantification systems. Therefore, for a given need, the operating parameters range has to be accounted for when choosing a system for a specific type of source. Overall, this justifies OGMP 2.0's recommendation to perform reconciliation between two different quantification methods, e.g., a source-level and a site-level one, to ensure a detailed and robust assessment of emissions.

## 5 Lessons learned and implications

We assessed performance of currently available quantification systems for midstream emissions based on 17 blind controlled release experiments. The controlled releases covered a wide range of situations, such as different flow rates (from 0.01 kg h$^{-1}$ to 50 kg h$^{-1}$), release heights (ranging from 1 m to 28 m), and different types of gas outlet shapes (e.g., open-ended, ring-shaped and linear). The analysis attempts to identify environmental and configuration factors limiting performance. Although the measurements were conducted under partially controlled conditions, low wind speed and unavoidable interferences between measurement systems have been identified as factors that affect measurement uncertainty.

Table 7 summarizes the findings of the present study. Most systems could report within an order of magnitude of the controlled release rate. Lidar 1 and Tracer have demonstrated average absolute errors below 50% on more than two-thirds of releases. The absolute errors of Lidar 2 and Tracer denote comparable uncertainties (Table 7). Drones specified relative higher uncertainties compared with other measurement systems. Self-reported uncertainties were not available for some systems. Available self-reported uncertainties determined as the standard deviation of a series of independent realization based on a theoretical calculation or a fixed value. Besides, for most providers, site-level measurement systems mean errors are higher than the self-reported uncertainties by technology providers.



Overall, the best performers are associated with deployment constraints. Lidar 1 requires the deployment of a helicopter. Although the present study did not investigate detection capability, Lidar 1 onboard helicopter lends itself to be integrated with routine pipeline patrolling. The mobile ground measurements (e.g., Tracer and Lidar 2) had difficulties accessing areas downwind of source emissions based on meteorological and road conditions. Tracer performs well if the acetylene release is well-collocated next to pre-identified leak areas and roads are available downwind. Lidar 2 had challenges positioning the

truck-based platform under certain wind conditions and could cover only 41% of releases. Lidar 2's sensitivity to wind conditions may be a less stringent limitation in real life than in this experiment. Here the time constraints of the experiment were fixed in advance and known to the participants. However, in real life, an operational application may allow for more relaxed time constraints and the ability to wait for favorable wind conditions, but not always. Lidar 2's small coverage of tests may imply implementation issues to perform measurements that cover a whole site, as they need to find appropriate

truck location depending on the location of the emissions and the wind direction, further research is needed on this. Ground-based measurements such as Fixed 1, Fixed 2, and OGI have limited detection distances. However, today no single technique that may be considered as a practical working standard for quantification.

**Table 7: A summary of findings from the present study.**

| Systems | Absolute errors (%) | Supplier specified uncertainty (%) | % of release where true emission rate is inside the uncertainty range | 0.5-2x (%) | 0.1-10x(%) | Release coverage (%) |
|---|---|---|---|---|---|---|
| Lidar 1 | 33 | N/A | N/A | 92 | 100 | 88 |
| Lidar 2 | 29 | 17 | 100 | 100 | 100 | 41 |
| Drone 1 | 140 | 55 | 20 | 40 | 100 | 71 |
| Drone 2 | 239 | 29 | 20 | 36 | 100 | 94 |
| Tracer | 19 | 20-30 | 92 | 92 | 100 | 82 |
| Fixed 1 | 175 | 13 | 25 | 50 | 100 | 82 |
| Fixed 2 | 84 | N/A | N/A | 78 | 100 | 53 |
| OGI 1 | 63 | 36 | 14 | 36 | 79 | 94 |
| OGI 2 | 74 | N/A | N/A | 25 | 69 | 82 |
| Hi-Flow | 80 | 12 | 0 | 33 | 100 | 29 |

The limited number of releases (17) implemented did not let significant influence emerge from wind speed and node shape.

Nodes clustered in Area B and its structure induced challenging conditions for single-node measurements during multiple-node releases, yet representative of mid-stream facilities. More controlled release experiments are needed to acquire more statistics, and test the dependence on a wider range of environmental parameters, especially wind conditions. Sensor precision may play a role in small release rates but was not demonstrated influence the releases above 0.1 kg h$^{-1}$ significantly. The fact shown in the present study that 0.01 kg h$^{-1}$ could not be quantified with any of these systems. There is an apparent

random character, mostly technique-dependent, not elucidated in the frame of the present study, but that could likely be





clarified with more data and comparing atmospheric turbulence and building configurations, and controlled gas temperature/injection speed and direction.

Additionally, the present study demonstrated that Tracer and Lidar 1 could be independently used for quantifying $CH_4$

emissions. However, their errors should be considered when performing reconciliation. Lidars and Tracer show better estimates (below 50%) of the total emissions among site-level measurement systems, and OGIs (ranging from 63% for OGI 1 to 74% for OGI 2) show stable and better estimates of the individual nodes during multiple-node releases compared with other source-level measurement systems. Therefore, Lidars/Tracer applied with OGIs together have the potential to obtain not only accurate estimates of total emissions but also accurate estimates from each node. Further work is needed to determine how these systems can be applied together to reconcile source-level and site-level quantification.

Only Lidar 1 appears to combine the advantage of site-level techniques and source-level precision, albeit at the cost and footprint of deploying a helicopter. Reliable site-level measurements are in principle useful to: a) identify all leak sources and b) provide a check for source-level inventories. Source-level measurements allow to rank fugitive emission sources, and to plan accordingly LDAR campaigns.

In the present study, the site was positioned in an environment selected for its isolation from other methane sources. In a

real-life context with nearby sources (e.g., industrial complex and proximity to agriculture), our assessment of node-level performance in single vs. multiple-node releases (Section 4.4) suggests that most measurement systems would see their performance degraded to some extent, depending on the proximity to external sources. With the influence of nearby external sources, the distinction between low-concentration and high-concentration regimes might not hold. The sensor precision would then be expected to play a role in the ability to discern specific plumes of interest from other nearby sources.

The ambitious OGMP 2.0 Level 5 reporting requires the complementary site-level measurements such as the ones scrutinized in our study. Level 5 is the highest grade and elaborates on top of level 4, a source-level estimate of asset emission based on measurements. The site selected for our study is considered an archetypal site of the natural gas midstream industry that would be using this reporting. Our study selected state-of-the-art systems currently available in Europe and able to perform measurements such as those required by OGMP 2.0 for the reconciliation process, for Level 5

reporting. In real-life applications, whether or not these measurement systems can fulfill the requirements of this reporting depends not only on individual technology performance, but also on the frequency of deployment and reconciliation methodology. However, we have shown that the definition of 'site-level' as considered in Level 5 reporting still represents a challenge for measurement systems. Indeed site-level's ability to distinguish individual sources is a bonus, as is source-level's ability to quantify at area level. Careful consideration of the integration of detection along with quantification would

be valuable, including standalone detection and quantification capability or using complementary detection sensors. This outcome should be taken into account when defining how reconciliation is to be performed. Level 5 requires reconciliation with the source-level estimate, which should be investigated in future research. Finally, we expect a continuous improvement of the accuracy of site-level estimates which may require such intercomparison to be repeated in the future.



*Data availability*. The data is not made publicly available in a repository, but can be requested from the corresponding author.

*Author Contributions*. YL performed data comparison and synthesis and wrote the manuscript. JDP, GB VBR and TMF designed the study. JDP and GB contributed to data analysis, synthesis and writing of the manuscript. MV, PB and JS contributed to advising, reviewing, and editing the manuscript. OW, JGH, CG, CD, CL, ER, YC, AF, JH, FI, NY, RA, HGN,
and EC contributed to instrument preparation, fieldwork and data analysis. DW and RH contributed to field supervision and technical guidance. MI contributed to data acquisition. AM contributed to operating the drone. JM contributed to preparation and field supervision. TE, SD contributed to data processing and data analysis. MU contributed to coordination and supervision. AS and SH contributed to data processing and data analysis as well as supervision of this study. FCM, ARB, PAR, AFF contributed to preparation, field work, supervision and data analysis. VBR, TMF and MZ contributed to
coordination, supervision and experiment design. RZ provided valuable feedback as a member of the Advisory Board.

*Competing Interests*. The authors declare that they have no conflict of interest

*Acknowledgements*. We would like to thank Nick Shepherd, Fritjof Biowski, Stefanie Ulbricht, Alberto Wahnon and Alfredo Badolato for their on-site operations and supervision for making the one-week experiment move smoothly; thank Luke Holehouse and Amy Paerson for the data analysis and data visualization. Additionally, we would like to thank pilots Andrew
Lake and Juergen Zimmer for performing the drone and helicopter flights on-site.

*Financial support*. This research was supported by GERG members and the project Eastern Mediterranean Middle East-Climate & Atmosphere Research Center (EMME-CARE) which has received the funding from the European Union's Horizon 2020 research and innovation programme under grant agreement No. 856612 and the Cyprus Government.

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
