# Peer review of "Assessment of current methane emissions quantification techniques for natural gas midstream applications"

_Atmospheric Measurement Techniques, 2023_

## Author Response (AR1)

We would like to thank the referee for his review and comments to improve the quality of this work. Following these comments, we have made major modifications and point-by-point responses. Please find our responses below (in blue color).

The manuscript describes the results of a large blind controlled methane release experiment. Ten different quantification techniques were assessed over an intensive release campaign, where emission rates were varied over several orders of magnitude in 2 hour blocks. More details on the setup of the measurement system are required including the location of the sensors (and reflectors) from the release point for each of the releases, details on the quantification method employed, elevation of the helicopter or drone flights, whether the technique was mobile or fixed during the experimental run, etc. This information could be included as supplementary information and is necessary for the reader make sense of the high errors.

Thank you for the general comments.

We provided a supplementary material to include more details of these measurement systems' technologies, measurement protocol, their quantification methods, field setup and deployment such as the location of the reflectors and cameras, specific meteorological/wind measurement systems, and the challenges during the experiment, etc. According to the details on the location of the sensors from the release point for each of the releases, it would hardly provide further insights on the results than the synthetic presentation of the current paper.

The study found high measurement errors across almost all techniques employed but the authors appear have omitted the obvious conclusion – 2 hours is far too short to accurately quantify a leak for many of the techniques employed.

In the past studies like Kumar et al. (2021 and 2022), Albertson et al. (2016), Brantley et al. (2014) and Foster-Witting et al. (2015) dedicated to the estimate of release rates from point sources using mobile measurements across the plumes and atmospheric dispersion models documented similar typical average precisions of 20%-30%. However, they relied on releases and shorter measurement time series lasting at least 20 min. Longer release durations (e.g. at least 30 min) would enable a much higher number of plume cross transects to be measured around the site, and this could ensure much more favorable inversion conditions. For the fixed ground measurement systems, it could be stationary over long durations (like days/weeks). In this study, we applied multiple measurement platforms including helicopter, drones, vehicles and fixed ground, therefore, we chose a median gradient duration 2 hours to conduct this controlled release experiment. Additionally, all the technology providers agreed on this duration and found the duration of their measurement slot sufficient except Lidar 2. Lidar 2 was clear since the very beginning that this timing was not appropriate for their technique. For helicopter, drones and cars, the duration of the tests is similar to the duration of their measurements in real conditions.

Having the majority of techniques able to estimate the emission within an order of magnitude (i.e. a long way from 10%) is not encouraging.

This study reports that Tracer, Lidar 1 and Lidar 2 were able to estimate good estimates (19%, 33% and 29% respectively).

More time is required to average the plume and optimize wind direction. This could potentially reduce the errors from approximately 100% and see them closer to 10%, as has been observed in comparable studies.

As mentioned above, past studies like Kumar et al. (2021 and 2022), Albertson et al. (2016), Brantley et al. (2014) and Foster-Witting et al. (2015) dedicated to the estimate of release rates from point sources using mobile measurements across the plumes and atmospheric dispersion models documented similar typical average precisions of 20%-30%. However, they relied on releases and shorter measurement time series. Bonne et al. (2023) reported good results (-69%-150% error range) for drones at Lacq, France. One critical point is that the average wind conditions could vary much in two hours, at some point, lengthening the duration of the release does not help better to get the average plume unless we could be in very optimal conditions of close to stationary wind over hours. In the real life, sources in industrial sites may hardly be stationary over long durations (like days/weeks). Besides, Caulton et al. (2018) recommended using at least ten plume cross transects to reliably constrain atmospheric variability and reduce the uncertainties in the estimation of the emission rates using mobile measurements. Additionally, the previous studies were not conducted at a real site. This is the added value of this work. There is real equipment interfering with the ideal dispersion of the plumes. Also, the providers did not indicate that more time was required to average the plume and optimize wind direction. We replicated as much as possible and, in all senses, real life operations, including with regards to duration of the measurements.

References mentioned above.

Albertson, John. D., Harvey, T., Foderaro, G., Zhu, P., Zhou, X., Ferrari, S., Amin, M. S., Modrak, M., Brantley, H., and Thoma, E. D.: A Mobile Sensing Approach for Regional Surveillance of Fugitive Methane Emissions in Oil and Gas Production, Environ. Sci. Technol., 50, 2487–2497, https://doi.org/10.1021/acs.est.5b05059, 2016.

Bonne, J.-L., Donnat, L., Albora, G., Burgalat, J., Chauvin, N., Combaz, D., Cousin, J., Decarpenterie, T., Duclaux, O., Dumelié, N., Galas, N., Juery, C., Parent, F., Pineau, F., Maunoury, A., Ventre, O., Bénassy, M.-F., and Joly, L.: A simultaneous $CH_4$ and $CO_2$ flux quantification method for industrial site emissions from in-situ concentration measurements on-board an Unmanned Aircraft Vehicle, Atmos. Meas. Tech. Discuss. [preprint], https://doi.org/10.5194/amt-2022-334, in review, 2023.

Brantley, H.L., Thoma, E.D., Squier, W.C., Guven, B.B. and Lyon, D.: Assessment of methane emissions from oil and gas production pads using Mobile measurements, Environmental Science & Technology, 48(24), 14508-14515, https://doi.org/10.1021/es503070q, 2014.

Caulton, D. R., Li, Q., Bou-Zeid, E., Fitts, J. P., Golston, L. M., Pan, D., Lu, J., Lane, H. M., Buchholz, B., Guo, X., McSpiritt, J., Wendt, L., and Zondlo, M. A.: Quantifying uncertainties from mobile-laboratory-derived emissions of well pads using inverse Gaussian methods, Atmos. Chem. Phys., 18, 15145–15168, https://doi.org/10.5194/acp-18-15145-2018, 2018.

Foster-Witting, T.A., Thoma, E.D. and Albertson, J.D.: Estimation of point source fugitive emission rates from a single sensor time series: a conditionally-sampled Gaussian plume reconstruction, Atmospheric Environment, 115, 101-109, https://doi.org/10.1016/j.atmosenv.2015.05.042, 2015.

Kumar, P., Broquet, G., Yver-Kwok, C., Laurent, O., Gichuki, S., Caldow, C., Cropley, F., Lauvaux, T., Ramonet, M., Berthe, G., Martin, F., Duclaux, O., Juery, C., Bouchet, C., and Ciais, P.: Mobile atmospheric measurements and local-scale inverse estimation of the location and rates of brief $CH_4$ and $CO_2$ releases from point sources, Atmos. Meas. Tech., 14, 5987–6003, https://doi.org/10.5194/amt-14-5987-2021, 2021.

Kumar, P., Broquet, G., Caldow, C., Laurent, O., Gichuki, S., Cropley, F., Yver-Kwok, C., Fontanier, B., Lauvaux, T., Ramonet, M., Shah, A., Berthe, G., Martin, F., Duclaux, O., Juery, C., Bouchet, C., Pitt, J., and Ciais, P.: Near-field atmospheric inversions for the localization and quantification of controlled methane releases using stationary and mobile measurements, Quarterly Journal of the Royal Meteorological Society, 148, 1886–1912, https://doi.org/10.1002/qj.4283, 2022.

The drones disturbing the plume profiles is also not helpful for the techniques that require estimates of plume shape.

It was mentioned in the study at Line 181-Line 183 that "The drones generate turbulences that can influence the structures of the plume measured by other platforms (in particular by Lidar 2), which can perturb the corresponding emission computation." It was okay for the rest techniques, and they had enough time for their measurements after the slot assigned to the drones. All the groups were informed in the first hour of each test that the drones were flying, and they were also informed by radio when the drones stopped flying.

I would encourage the authors to consider including ensemble estimates (i.e. averaging multiple techniques) for the emission rate. This may yield a better estimate for the blind releases and it would be interesting to see how this could be optimised for the minimum number of the techniques to achieve a reasonable emission rate estimate.

As shown in Figure 4, averaging ensemble estimates of multiple techniques could optimize the emission estimates (for example, Drones overestimated the emission rates and OGIs underestimated the emission rates). In the implications part (Section 5), we highlighted that "Therefore, Lidars/Drones/Tracer applied with OGIs together have the potential to obtain not only accurate estimates of total emissions but also accurate estimates from each node. Further work is needed to determine how these systems can be applied together to reconcile source-level and site-level quantification." at Line 500 to 505.

Line 55 - please include a statement about the short measurement time and its potential influence on the error through insufficient dispersion modelling.

Please refer to our response to the general comments. Refer to the statement "However, in real life, an operational application may allow for more relaxed time constraints and the ability to wait for favorable wind conditions, but not always." at Line 531.

Line 61 - remove significantly - it depends on the technology used to burn gas or coal.

"Significantly" was removed.

Line 106/107 - is the underground gas storage facility defunct, i.e. not working? I was under the impression this is the source of the methane used in the experiments.

The underground gas storage facility is not working, and it was not the methane source in the experiment. Section 2.2 Controlled release facility includes the detailed description of the methane source. The experiment was conducted at the compressor station of a defunct underground gas storage facility to provide a realistic environment for such measurements.

Line 125 – Can the authors clarify the source (and purity) of the methane used in the experiments? Presumably it is from the nearby underground storage site but it is not explicitly stated anywhere.

Section 2.2 Controlled release facility describes the source of the methane used in the experiment, and "the gas used for this experiment was 99.95% by volume pure methane, supplied by a commercial pressurized gas cylinders provider." was added between Line 143 and Line 145.

Line 126 - The compressor is not in …

The sentence was rewritten as "A mothballed compressor station located in Spain was selected as the test site. The compressor station has various compression equipment for injecting and treating gas extracted from a nearby underground gas storage (Fig. 1)."

Line 167 - "… across the releases a threshold will emerge between … with instrumental accuracies) and "high concentration …."

The sentence was modified.

Line 174 - not sure what this short sentence means. Please clarify.

The short sentence "We used a relatively uniform sampling of the emission rates to infer it." at Line 174 was removed.

Line 185/186 - a single anemometer at 5m could be contributing to the significant errors for many of the techniques. It would be have been preferable to measure the wind properties at multiple heights, particularly given the large range in release heights (1.5 to 28m).

The wind data from this anemometer was used for the analysis shown in Fig.5. However, the different emission quantification systems relied on different types of wind measurement or source

of information on the wind during the campaign following pre-established protocol. All the details regarding the different wind measurement protocol and instruments were added to the supplementary material.

Line 191 - please include the release heights with the Node labels for easy reference.

The heights were added to the Node labels in the table.

Line 226 – Table 2. The quantification algorithm column does not adequately describe the techniques employed for the measurements, providing the reader little confidence in the techniques employed. Please include a more fulsome description in this table and as supplementary information (or link to relevant published techniques). For example, do the OGI techniques use wind direction/speed information? Is LIDAR 2 mobile or fixed? Additional information required is the location of the sensors and reflectors, were the techniques mobile or fixed during the 2hr experiment, did the location of the measurement equipment change for each experiment, how far away where the handheld, fixed sensor, truck, drones, etc from the release points? A more useful summary could be included in the table with the measurement setup for each technique described in the supplementary information.

To improve the description of the emission quantification systems, the supplementary material was updated by including more information about the employed measurement systems' technologies, field setup and deployment such as the location of the sensors and the reflectors, the emission quantification methods and tools, the specific meteorological/wind measurement systems, and the challenges they had during the field campaign. In addition, some plume mapping examples measured by these technologies were added into the supplementary material. According to the details on the location of the sensors from the release point for each of the releases, it would hardly provide further insights on the results than the synthetic presentation of the current paper.

Line 249 - "This consideration is essential when evaluating …"

The sentence was modified.

Line 253 – Please expand on this - 2 hours is a short window for the experiments and would rule out several (and possibly more accurate) quantification techniques, i.e. most techniques that require plume simulation. The sensors may not be in the optimal location downwind of the release point during the 2 hour period or experience a range of wind directions (i.e. null results) that would be helpful for Bayesian analysis. Such an approach could significantly improve the emission rate and error.

Please refer to our response to the general comments. The average transport conditions can vary significantly in two hours, therefore, longer measurements for the releases may not help getting the better "average plume", which would correspond to steady wind conditions.

Line 270, 290 – the plots in Figure 3 and Figure 4 show that many of the techniques had large errors – higher than observed in similar studies. This is most likely due to the short timeframe of the experiments. The tracer (and LIDAR) techniques stand out as the most accurate techniques by

far. Why is this? Are these techniques not dependent on describing the shape of the horizontal plume?

Tracer applied around 45 min during each release test to estimate the emission, and they reported that for most tests they properly caught the plumes at the downwind. Lidar 1 based on the helicopter scanned most plumes during the campaign to estimate the total emissions as mentioned in their report. Lidar 2 based on the truck had difficulties finding a proper location to catch the plumes and interfered by drones for some releases, but when they were downwind the single or ensemble of plumes, they managed to catch good estimates of the total emissions within the two hours. The average atmospheric conditions can vary significantly in two hours, therefore, longer measurements for the releases may not help getting the better "average plume". However, the revised manuscript now mentions the possibility to expand release duration to refine analysis per technique (Line 545).

Line 320 – The errors are not correlated with time over the 2 hour period, but given longer periods of measurement time (e.g. days/weeks) some of the techniques would have significantly lower errors for the emission rates.

From some individual techniques' perspectives this may be true, but not for all. And for some technique there is a significant cost to longer duration. Moreover, intermittency of emissions is a frequent feature in the industry covered here and others. If emission location and intensity are not stationary over the targeted durations (days/weeks), we question whether such a long duration is relevant.

Line 325 - internal review comment?

Sorry, these were internal comments. The sentence was removed.

Line 326 - replace didn't with did not.

This was modified at Line 326.

Line 341 - Can't read y-axis text at bottom of plot. What does this represent? Please describe in the figure caption.

The description of y-axis "y axis ranging from 0 to 100%" was added in the figure caption.

Line 346 - the role of the wind needs a more thorough consideration, e.g. how it affects plume shape estimates for some of the techniques, measurements not at the right height, etc

Each quantification technique used its own predefined meteorological measurement system or meteorological condition characterization, and its own way of using wind/meteorological data to derive emission estimates according to its predefined quantification protocol. For example, Lidar 2 used a 12 m fixed metrological mast with four channels at different heights. Above information about wind's role in emission estimates for the technologies was added to the supplementary material.

Line 354 – "do not reveal any clear relationship between the wind speed and direction and errors". Given the errors are so high, I'm not sure this is an appropriate conclusion.

Many of the results could be described as acceptable or good (below 100%), and we do not see any relationship between the wind and the best estimates.

Line 360 – improved wind measurement protocols would be very beneficial.

This is a statement that we fully endorse. Wind measurements differed across the techniques and could generally be improved.

Line 374 - Table 4. This is not particularly useful - remove.

Since the gas release outlet points' share were different including open end, ring shaped and linear releases, Section 4.3.4 aimed to determine if the different type nodes would affect the errors in quantifying the emission rates. Therefore, Table 4 here calculated the absolute errors based on different nodes' release situations, which is necessary here to robust the description for this section.

Line 396 - Figure 6 can be removed - it is virtually identical to figure 4.

Figure 6 aimed to show the technologies' performance excluding Node 1, which was a challenging emitting node set at the height of 28 m. It would be more visible to describe and compare with Table 5.

Line 424 - I'm not sure you can make this conclusion given the very high errors, i.e. performance is not good for either.

There are good estimates from three techniques (Tracer and Lidars) and there are also good estimates at node-level from specific measurement systems, therefore, we can get these conclusions for specific systems, discarding the poorer performers.

Line 458 - "Most systems can report within an order of magnitude of the controlled release rate". This not an encouraging outcome, given the experiment was under constant flow rate and controlled conditions. The authors need to reframe their results. Why are the errors so high and what recommendations can be made to reduce them? Not having enough measurements at optimal wind orientations (enough time) and a good model of wind dispersion are likely to be key issues.

Please refer to our response to the general comments. These are commercial and marketing available measurement systems based on current technologies. They developed their own quantification methods or tools as would be done for a commercial service. To evaluate these technologies' performance, this study brought them to the blind controlled release experiments "as-is".

Besides, during the field campaign for some releases, the interference between different systems degraded a few results. For example, Lidar 2 knew they would exclude the measurements when the drones were flying. All these details and information like measurement protocol, their

quantification methods, field setup and deployment such as the location of the reflectors and cameras, specific meteorological/wind measurement systems, and the challenges during the experiment were added to the supplementary material for each participating measurement system. One aim of this study is to assess performance in real life conditions to provide gas operators with information on this when the technologies are used.

We would like to thank the referee for his/her review and comments to improve the quality of this work. Following these comments, we have made major modifications and point-by-point responses. Please find our responses below (in blue color).

Liu et al. assessed 10 available methane emission quantification techniques in a series of controlled release experiments covering a range of different emissions rates. The performance has been evaluated from multiple angles, which is very informative and may be useful if the results are convincing. Overall, I feel that essential information on how different methods work should be provided to make any conclusions convincing. This type of controlled release experiment work is highly desired and could potentially be very useful. The topic is very suitable for AMT, and I can recommend publication only after the following concerns have been addressed.

Thank you for the general comments. We provided a supplementary material to include more details of these measurement systems' technologies, measurement protocol, their quantification methods, field setup and deployment such as the location of the sensors and cameras, specific meteorological/wind measurement systems, and the challenges during the experiment, etc.

4.1 some analyses beyond the apparent results will be appreciated, e.g., what are the slopes of the linear fits? And the uncertainties of the slopes? How many measurements have been made for each methodology? In terms of time, flights, etc.

The slopes and the uncertainties of the slopes were added to the figure. And the statement "Lidars, Drone 1, Tracer and OGI 1 had relatively high R-squared (above 0.8)." was added to Line 291. More information such as these technologies' field measurements, deployment and strategies were added to the supplementary material. In addition, some examples measured by the technologies were also added to the supplementary material.

4.3.2 Dependence of error on emission rate, it is good to see the quantification limit. The question is whether the relative error depends on the emission rate. This is so far missing.

We have Section 2.3 Test scenarios and organization of the experiment and then Section 4.3.2 dependence of error on emission rate about this.

4.3.3 Role of wind: one could check whether the relative errors depend on wind speeds, particularly for those methods that are based on atmospheric dispersion.

We have Section 4.3.3 discussing the role of wind, from Line 355 to Line 370.

5. Lessons learned and implications are too long. This section should be a concise one, and should not bring up new summaries and discussions.

We agree with the suggestion. Section 5 lessons learned and implications was modified accordingly, and also a short conclusion section was added.

Some detailed comments:

L59: billion standard m3 per year?

Yes, here is billion standard $m^3$ per year.

L94-97: break the sentence into two.

The sentence was rewritten into two sentences "With the rapid development of current technology, Sherwin et al. (2021) have shown that an airplane-based hyperspectral imaging $CH_4$ emission detection system can detect and quantify over 50% of total emissions from super-emitters. Super-emitters are fewer than 20% of sources while contributing more than 60% of total emissions (Duren et al., 2019)."

L98: what are the recently regulated venting limits? It will be helpful to provide here.

An example was added "for example, regulations in Alberta, Canada to limit the methane emissions from site venting 12.3 kg $CH_4$ $h^{-1}$" at Line 98.

L147-150: It is confusing here. What systems were calibrated by the NPL? What are the uncertainties of the CRF and MidiCRF?

It is modified that "The CRF and MidiCRF were calibrated by NPL". The uncertainty of the CRF and MidiCRF are shown in Figure 5 the lower panel.

Figure 2: The quality of the figures is too low to be read.

The figure was replaced with an improved quality and uploaded as a separate file.

L204-206: More descriptions of the techniques to quantify the CH4 emissions are indispensable, especially the precision and the stability of the measurements of various systems.

More description of the techniques and their quantification methods were added to the supplementary material according to the reviewer's comment.

L215-218: It has been said before at L205-206. Hi-Flow has been used in previous studies. It will be needed to provide more details here, such as the flow rate and the technique used to determine CH4 concentrations.

The description of Hi-Flow technique was added to the supplementary material. And Hi-Flow used in this study is a prototype which had not been used before. The sentence at Line 218 was modified as "it was included in this study to assess its performance for fugitives' quantification".

L234: the abbreviation of absolute error for "the absolute value of the relative error" is quite confusing. Better relative error or something else.

Relative errors in absolute values were used in this study, so here is not relative error. To be specific of this study, we chose "absolute error" as the abbreviation.

L286: Please rewrite the sentence "Since the release they could not measure the appropriate nodes were excluded from the analysis"

The sentence was rewritten as "since they excluded the results for the release tests the emitting nodes not caught".

L300 Any significant deviations when wind speed is smaller than 2 m/s?

Section 4.3.2 discussed the role of wind, as mentioned at Line 363 Low wind speed values (below 2 m s$^{-1}$) and specific wind direction sectors prevented some participants from providing valid estimates during specific releases. However, considering the valid estimates shown in Figure 5, it is not clear that releases for wind speed at 2 m s$^{-1}$ or less are worse.

L325: see comment, which comment?

Sorry, these were internal comments. The sentence was removed.

L328-330: This is very ambiguous. Please be specific about what techniques are referred to.

The sentence was rewritten as "for the techniques Drone 1, Drone 2, and Lidar 2, increasing time coverage can also improve the ability to cover all nodes".

---

## Author Response (AR2)

We would like to thank the referee for his review and comments. Following these comments, we have made minor modifications and point-by-point responses. Please find our responses below (in blue color).

Minor review comments and suggestions:
The addition of the new supplementary information is huge improvement and provides greater confidence in the study. The authors may have misinterpreted my comments regarding why short measurement times can be problematic. It's not so you can better average the plume, but that you have more opportunity to take measurements at optimum wind conditions and orientations relative to the source. Plumes swing around and, depending on the sampling rate and location of the measurement system, you may only catch a few optimal measurements which leads to higher quantification errors. Longer measurement times mean you have a higher chance to collect more measurements from the right wind direction and at a good wind speed. You can extend this further using Bayesian inversion analysis, using non-perturbed measurement information with wind orientation (from "bad measurements") to better constrain the width (hence shape) of the plume under different wind speeds, directions and mixing conditions. This provides more information to generate an accurate estimate, but it takes time (more than 2 hours). As highlighted in the supplementary information, unfavourable wind speeds and directions were an issue for several of the measurement techniques. A comment reflecting this would be useful.

Thank you for the comment. We added the statement "Several of the monitoring techniques would have benefited from longer release durations with longer measurement windows to yield more accurate estimates. However, some specific techniques lacked measurements or favorable measurement conditions during the 2-hour releases." to Line 444.

Ln 194 – (units in kg/h-1, node height also shown in header)

The unit ($kg\ h^{-1}$) was added to the table header.

Ln 221, table 2-please include general quantification approach for OGI1 and OGI2 for consistency.

The quantification approach description for OGI1 and OGI2 was added to Table 2.

Ln 227, Figure 3. Regression slope for Fixed 1 doesn't appear to go through zero. Please update slope and r2.

The x axis and y axis were rearranged to make 1:1 line clearer.

Ln 331, Figure 5 – Mean error values in red on plot are too small to read.

Figure 5 was updated using bigger labels.

Ln 337, Figure 6 – I still suggest that this figure is unnecessary and simple comment that removal of vent stack emissions did not appreciably change the errors for the different techniques is sufficient, but I'll leave that to the editor!

Figure 6 was moved to the supplementary material.
Ln 445, removes Besides.

"Besides" was removed.

Ln 469, do you mean source-level precision or accuracy?

It is source-level precision.

Ln 527 – It appears that the tracer technique and helicopter based Lidar are the most reliable and effective techniques for accurate quantification of emissions, provided that it is possible to release the acetylene at the correct location for the tracer test (consistent with what was observed at the Ginninderra controlled release experiment). The tracer doesn't require plume modelling and the helicopter technique is not constrained by on the ground infrastructure. I suggest that this is quite an important finding from this study for this type of application.

The statement "The tracer and helicopter-based Lidar appear to be reliable and effective techniques to quantify the emissions accurately. The tracer does not require local dispersion modelling and the helicopter technique is not constrained by the ground infrastructure. However, the tracer technique could be hampered by the lack of ability to locate the tracer release appropriately to ensure that the tracer atmospheric dispersion reflects the methane one (Ars et al., 2017)." was added to Line 496.

Supplementary information comments:

Section 1, ln 1 – replace Aera with Area

It was corrected.

Section 1.1, ln 6 – Area B

It was corrected.

Section 1.6, ln 8 – Area A

It was corrected.

Section 1.7, ln 11 – athermalize is the incorrected word. Equilibrates?

It was replaced by equilibrates.

Section 1.8 – please describe the type of measurements system for the methane detector (e.g. IR?)

"A methane detector (Huberg Laser One based on Tunable Diode LASER Absorption Spectroscopy)" was added to Section 1.8 at Line 2.

Section 1.9 and 1.10 – please describe in general terms the approach taken to quantify the emissions, e.g. inverse Gaussian plume modelling using meteorological data?

The quantification approach description and relative websites about the quantification software were provided to Section 1.9 and Section 1.10.